# The effect of school smoke-free policies on smoking stigmatization: A European comparison study among adolescents

**Pierre-Olivier Robert**[1]*, **Adeline Grard**[1], **Nora Mélard**[1], **Martin Mlinarić**[2], **Arja Rimpelä**[3,4], **Matthias Richter**[2], **Anton E. Kunst**[5], **Vincent Lorant**[1]

**1** Institute of Health and Society, Université Catholique de Louvain, Brussels, Belgium, **2** Institute of Medical Sociology (IMS), Medical Faculty, Martin Luther University Halle-Wittenberg, Halle (Saale), Germany, **3** Faculty of Social Sciences (SOC), Unit of Health Sciences, Tampere University, Tampere, Finland, **4** Department of Adolescent Psychiatry, Pitkäniemi Hospital, Tampere University Hospital, Tampere, Finland, **5** Department of Public Health, Academic Medical Center, University of Amsterdam, Amsterdam, The Netherlands

* pierre-olivier.robert@uclouvain.be

**Data Availability Statement:** There are ethical restrictions on sharing data which contain potentially personally identifiable information pertaining to adolescents. Information about the

## Abstract

The increasing denormalization of smoking by tobacco control policies and a normative smoke-free climate may shift power towards adolescent non-smokers. It is unclear, however, how common stigmatization of smokers is among adolescents or how stigmatization relates to the denormalization of smoking in their school and social environment. This paper aims to measure (1) whether stigmatization among European adolescents varies according to smoking status and socioeconomic position (SES), and (2) whether stigmatization is greater in school environments in which smoking is denormalized (i.e. those with low smoking rates and strong school tobacco policies). Data on 12,991 adolescents were collected in 55 schools in seven European countries (SILNE R-survey, 2016/17). We applied Stuber's adapted scale of perceived stereotyping and discrimination towards smokers to smoking status and five variables indicating a power shift towards non-smokers: the school's tobacco control policy (STP) score, the percentage of adolescents in the school who smoke, parents' level of education, students' academic performance, and the percentage of their friends who smoke. Multilevel regressions were applied to the global score for perceived stigmatization. Discrimination against smokers and stereotyping of smokers were frequently reported. Smokers reported less 'perceived stigmatization of smoking' than non-smokers (Beta = -0.146, p < 0.001). High-SES students reported stereotyping and discrimination more frequently than lower-SES students. The perception of stigmatization was lower among students whose academic performance was poor (Beta = -0.070, *p* < 0.001) and among those who had friends who smoked (Beta = -0.141, p < 0.001). Stigmatization was lower in schools with greater exposure to smoking and was not associated with the school's STP score. Perceived stigmatization of smoking is common among European adolescents. Smokers themselves, however, perceive stigmatization less often than non-smokers. Strong school tobacco policies do not increase stigmatization, but a social environment that is permissive of smoking decreases perceived stigmatization.

ethical and legal restrictions is included in the manuscript. All study participants were asked to sign an informed consent agreement prior to participating in the study. Following the recommendations of the European Commission, national and local organizations, the datasets generated in this study are not publicly available. Making all the data available would contravene the Consortium Agreement that was developed and signed before the start of the SILNE-R project, specifically with regards to the ownership and use of the project output. Furthermore, we would like to retain control over the use of the data in order to avoid it being used inappropriately by the tobacco industry. In 2021, however, any person can apply for access to the data through the SILNE R consortium (the data owner). The contact person, as data holder, is Anton E Kunst, a.e. kunst@amsterdamumc.nl. Within the Institute, the two contact persons are: Regina Below, from the research department, regina.below@uclouvain.be; and Alaa Mahboub, for technical support, alaa.mahboub@uclouvain.be.

**Funding:** This study is part of the project SILNE-R 'Enhancing the effectiveness of programmes and strategies to prevent smoking by adolescents: a realist evaluation comparing seven European countries', which was supported by the European Union's Horizon 2020 research and innovation programme under the SILNE-R Grant Agreement number 635056. URL: https://ec.europa.eu/programmes/horizon2020/en Coordinator: AEK The funders had no role in study design, data collection and analysis, decision to publish, or preparation of the manuscript.

**Competing interests:** The authors report no conflict of interest in relation to this study.

## Introduction

Denormalization strategies, designed to influence social norms in order to promote reduction of smoking in the society [1], have been developed and supported by numerous public health organizations, such as the Centers for Disease Control (CDC) and the World Health Organization (WHO) [2]. These strategies restrict smoking in public places, limit access to tobacco products through the prohibition of sales, and set regulations for tobacco advertising. They can also include media campaigns that raise awareness of the dangers of passive smoking [3]. One of the underlying assumptions on which they operate is that smokers who experience the stigma of smoking are more likely to quit than those who do not [4]. As a consequence of the denormalization of smoking, the general public has developed a negative attitude towards smoking [3, 5, 6].

Stigmatization occurs when a person is labelled as deviant and when negative stereotypes cause stigmatized individuals to be separated and discriminated against in an environment in which they have less power than the non-stigmatized [7, 8]. Power stems from a difference in social capital or socioeconomic position and from the political regulation of behaviour. In the specific case of smoking, such an unequal power dynamic may be strengthened by the increasing regulation of smoking and the low socioeconomic position of smokers [3, 9–12]. Researchers, however, have different opinions on whether smokers are stigmatized [9]: while stigmatization may imply deviation from a norm, not all deviations from the norm lead to stigmatization [8]. Given the upscaling of school tobacco control policies (STPs), it remains unclear whether young smokers feel stigmatized.

Although the stigmatization of smokers has been investigated in adults, there has been little quantitative research on smoking status and denormalization in adolescents. Understanding adolescent stigmatization in relation to smoking would be valuable for two reasons. First, adolescents are increasingly exposed to smoking denormalization at school, with the rise of school smoking bans, and outside of school, in settings that are key to their socialization, education and wellbeing. Second, stigmatization is associated with other risky behaviours, which affect both mental and physical health. As smoking remains more prevalent among adolescents from lower socioeconomic backgrounds, it may be a factor that increases socio-economic inequality in health [13].

### Theoretical background of stigmatization

Stigmatization depends on access to social, economic, and/or political power [7]. It is worth noting that the early development of non-smokers' advocacy groups coincided with an increase in the concentration of smoking within populations of lower socioeconomic status (SES). As a result, it may have become easier to stigmatize smoking as an undesirable and deviant behaviour [3, 14, 15]. The rise in the stigmatization of smokers and the increasing concentration of non-smokers in groups of higher socioeconomic status may not, therefore, be independent of one another.

As smoking bans have become more common, the social distance and physical separation between smokers and non-smokers has increased. This reduces their opportunities to interact socially and their likelihood of doing so. In some ways, this separation has been entrenched by TCPs, which have pushed smoking, and therefore smokers, out of public spaces, increasing their social exclusion and their sense of being outsiders [16–18]. That separation may also occur in adolescence. By promoting early experience of the stigmatization of smoking at school, STPs may also lead to a 'conscious and deliberate act' of separating smokers and non-smokers [19]. STPs denormalize smoking by reducing the visibility of the behaviour, thereby marginalizing smokers [20].

The stereotyping of smokers may also be a consequence of the denormalization of smoking. Stuber et al. showed that 38% of smokers who were interviewed reported feeling that, overall, people 'think less' of someone who smokes [21]. This negative image is associated with undesirable characteristics that reduce a person's status in the eyes of the people who stigmatize them [6, 7, 21–23]. Such stereotypes may also apply to adolescent smokers [24–26]. Given that TCPs lead to the denormalization of smoking, stereotyping and discrimination are inevitable corollaries of prevention [27]. To date, however, very little literature has examined the association between STPs and stigmatization in adolescents.

Discrimination, another consequence of the denormalization of smoking, implies a loss of social status [7, 27]. Research shows that some adult smokers report discriminatory treatment by healthcare services, such as being placed lower on waiting lists [3], or in the labour market [5]. Because of the enforcement of STPs, some adolescents may experience discrimination at school due to their smoking status. There is, however, little literature on the stigmatization of adolescent smokers.

Smoking bans and smoke-free culture shift the balance of power towards (young) non-smokers. Unlike most adults, however, adolescents are engaged in building their own identities; they may pay more attention to how others see their behaviour and they may be more receptive to a negative smoking climate [28]. Adolescents who smoke may therefore experience a higher degree of stigmatization than adults [29, 30].

In summary, as smoking becomes increasingly denormalized, smokers may be vulnerable to stigmatization. There is, however, very little quantitative evidence of such a link, and very few studies that compare adolescents in different contexts of smoking regulation. Given that the overall denormalization strategy requires the participation of schools, it is important to know whether stigmatization applies to young smokers at school.

In seven cities with tobacco control policies of varying degrees of strictness, we aimed to measure [1] whether European adolescents perceive a stigmatization of smokers and whether that stigmatization varies according to smoking status and socioeconomic position (SES), and [2] whether stigmatization is greater in school environments in which smoking is denormalized (i.e. those with low smoking rates and strong school tobacco policies).

## Materials and methods

This research was based on the 'SILNE-R' study, which was conducted in 2016–2017 in seven European cities. In each city, schools were selected from the local register of schools and were categorized as either high- or low-SES schools. Six to twelve schools in each city were selected and participated in the SILNE R survey. The schools were stratified according to information that was specific to each country: either the type of school (Italy, Germany, and the Netherlands) with (non-)academic tracks, the socio-economic ranking of the school by the educational authorities (Belgium and Portugal), or the area's socio-economic status (Ireland and Finland). Data were collected in 55 schools from 12,991 adolescents. In each school, two groups of school students were selected from grades corresponding to the ages of 14 to 16. In those two grades, all registered adolescents were invited to participate in the survey. The participation rates among adolescents were above 84% in Namur (Belgium), Amersfoort (the Netherlands), and Tampere (Finland). They were 80% in Dublin (Ireland), 79% in Latina (Italy), 76% in Coimbra (Portugal), and 66% in Hanover (Germany). In Germany and Italy, active parental consent was required, which resulted in a lower response rate than that in the cities where passive consent sufficed (See S1 Table for more information on the response rates.). Adolescents registered at these schools were invited to complete a written survey about their social relationships at school, their health behaviours, rules about smoking, and their

sociodemographic characteristics. Adolescents were considered non-participants if they were absent on the day of the study or unwilling to participate and were excluded if their questionnaires were blank or seemed too incoherent.

In each city, ethical approval was obtained from (sub-) national or local organizations, for which full references are provided (See S1 File). In some cities, permission to conduct the survey was also requested from the educational authorities. In accordance with each country's regulations, school principals, parents, and adolescents received leaflets, information letters, and parental consent letters. All study participants were asked to sign an informed consent agreement prior to participating in the study. The questionnaire for adolescents was translated into local languages from an original written English (see S2 File) version and was piloted in one school in each country. A few questions were reformulated following this piloting of the questionnaire in order to make them more understandable. The schools where the questionnaire was piloted were excluded from the sample of schools chosen to carry out the survey afterwards and their data were deleted. A detailed description of the design and concepts of the SILNE study has been published elsewhere in relation to the first wave of this study [31].

## Measurements

The main outcome variable, overall perceived stigmatization of smokers—in other words, the broad perception of a socially imposed stigma upon smokers—was captured using a standardized survey (see S2 File) and was measured using a short version of Stuber's scale of stigmatization (devaluation index). It is made up of four items. Two concern stereotyping: (a) *most people think less of people who smoke*; and (b) *most people believe that smoking is for losers*. The other two concern discrimination: (c) *most non-smokers would be reluctant to date someone who smokes*; and (d) *most non-smokers would not hire a smoker to babysit their children* [32]. The replies (from *strongly agree* to *strongly disagree)* were rated from 0 to 3. An average stigma score was calculated by adding together all scores for the four Likert items and dividing by four (mean = 1.62, std = 0.55). The score was then log-transformed to reduce skewness and facilitate parametric tests such as linear regression.

To control for the association between behaviour and stigmatization, we categorized the smoking variables of each stage of smoking status, from initiation to daily use, into six groups: non-smokers, people who had tried smoking at least once, occasional smokers (experimenters), weekly smokers, daily smokers, and ex-smokers [31]. This categorization helped to show how different experiences of smoking are associated with reaction to stigma.

We then identified three smoking environment variables in which power may be shifted between non-smokers and smokers. The first was school tobacco policies, which operated across three dimensions: comprehensiveness (does the STP apply everywhere and to everyone?), enforcement (was the STP actually enforced, and were sanctions associated with breaking the rules?), and communication (was the STP formally communicated by schools and, if so, by what means?). To quantify this variable, the perspectives of adolescents and members of school staff were used to create a school tobacco policy score for each school. In this study, between 24 and 44 school staff members participated in each city. We administered the questionnaire to at least three staff members per school, including, where possible, someone from the management, someone from the teaching staff, and someone in charge of health and prevention matters. Ranging from 0 = 'low' to 10 = 'high', this score indicated the extent to which the policies were perceived as being comprehensive and enforced, and properly communicated. We have provided, in S2 Table, a detailed list of all items of this STP score. Overall, staff and student perceptions of STPs were significantly correlated (r = 0.46, p = 0.0034). Their scores were also significantly correlated for policy comprehensiveness (r = 0.55, p = 0.0003)

and for policy enforcement (r = 0.51, p = 0.0010) [33]. The second variable was the proportion of weekly smokers in each school. The third was weekly smoking among peers, which is a potential indicator of the support a smoker can receive in his/her network to continue the behaviour.

As the SILNE-R study was a social network study, all students were asked to nominate up to five schoolmates they considered to be their best and closest friends. For each adolescent, we computed the proportion of smokers in the first out-degree separation set of friends [34].

As explained above, stigmatization depends on the distribution of economic and political power. We therefore identified two individual socioeconomic position variables that might shift power among adolescents at school, and that might also predict future socioeconomic status (SES). On the basis of each country's education system, parents' level of education was classified as low, medium, high, or unknown. Adolescents' academic performance (low vs high) was based on the students' self-reported marks from the previous year: 'Which of the following best describes your marks last year?'. This variable included five initial values based on each country's academic performance assessment system. Those values were dichotomized into two categories: low and high.

## Confounders

As TCPs differ between countries in the European Union, we used the country as a confounder. Countries were ranked according to the Tobacco Control Scale (TCS) developed by Joossens and Raw, ranging from high scores (Ireland-70 and Finland-60) to medium (the Netherlands-53, Italy-51, and Portugal-50) and low scores (Belgium-49 and Germany-37). The TCS includes factors such as taxes, restrictions on smoking in public places, consumer information (e.g. public information campaigns, media coverage, publicizing research findings), comprehensive bans on the advertising of all tobacco products, labels with health warnings on tobacco products, and treatment to help dependent smokers to quit. The scale allocates points to each policy, with a maximum score of 100: prices 30, smoke-free public places 22, spending on public information campaigns 15, comprehensive advertising bans 13, large health warnings 10, cessation support (treatment) 10 [35, 36]. However, the TCS score does not take school smoking bans into account. That is why we created a school tobacco policy score for each school for our analyses (see S2 Table) [33]. We also controlled for age and gender.

## Data analysis

After excluding 279 observations due to missing data, we first tabulated the stigmatization items, which were dichotomized into two categories: agree and disagree, according to the adolescents' smoking status and other covariates. A chi-2 test was included in this analysis. To quantify the association between smoking status, individual socioeconomic position variables, smoking-environment variables, and stigmatization, we then performed multilevel linear regressions, using the school as a random intercept. Due to the hierarchical structure of the data, observations were not independent. We therefore used multilevel models. The school was included in our models as a random intercept to take into account the correlation between different observations for a single school [37].

For linear regression, we used the log-transformed stigmatization score. We also categorized smoking status into three different groups: non-smokers, including ex-smokers and those who had never smoked, occasional smokers, including those who had tried smoking at least once and experimenters, and weekly smokers, including daily and weekly smokers. In Model 1, using the school as a random variable, we regressed the score of the stigma for each variable, and controlled for gender, country, and age. In Model 2, we added two individual

socioeconomic position variables (father's and mother's education and academic performance) and three smoking environment variables (the percentage of adolescents who smoked weekly per school, the proportion of weekly smokers among peers, and the school tobacco policy (STPs) scores. We stratified the previous analysis by smoking status.

Finally, stigmatization is likely to follow a different pattern in a school with very few smokers than in a school where smoking is rife. Indeed, as shown elsewhere [38], the gradient of smoking across academic performance is much more pronounced in schools with a higher SES than in those with a lower SES. We therefore tested the interaction of Models 1–2 with the school type, using the national classifications of schools' SES [31]. Statistical analyses were carried out using SAS 9.3.

## Results

The overall perceived stigmatization of smoking was greater for the items related to discrimination against smokers: 'would not hire a smoker to babysit their children' (78%) and 'would be reluctant to date a smoker' (56%) (Table 1). While there was a high overall perception that smokers were stereotyped, with over half of adolescents agreeing that people thought less of a smoker (53%), only 31% of students agreed that people believed smoking was for losers.

Although overall perceived stigmatization of smokers was associated with smoking status for all items, this was truer for discrimination items than for stereotyping. Fewer daily smokers than non-smokers reported that '*a non-smoker would not date a smoker*' (61.2% *vs* 41.9%). Furthermore, the more experience a respondent had of smoking, the lower the overall perceived stigmatization of smokers was for all items. For example, adolescents who had quit smoking reported less stigmatization than non-smokers, but more stigmatization than daily smokers.

STPs had a significant ∩-shape, but a moderate association with smokers' stigmatization. Permissive and strict STPs were associated with slightly less stigma than STPs that were implemented to an intermediate level. There was a marked and significant trend between exposure to smoking and perceived stigmatization of smokers by others: the more frequent smoking was in the school and among best friends, the lower was the perceived stigmatization of smokers by others. The perception of smokers as losers was more common in schools with a very low prevalence of weekly smoking than in the schools with the highest prevalence (44.2% *vs* 21.5%, $Chi^2$ = 750.3). Similarly, the statement 'most non-smokers would not hire a smoker to babysit their children' was agreed with less by those who had a high proportion of friends who were weekly smokers ('$\geq$ 50% of friends who smoked', 65.8%) than by those who had no friends who smoked ('no friends who smoked', 81%; $Chi^2$ = 188.2). There was less perceived stigmatization of smokers among low-SES adolescents (as expressed either by school performance or by parental education). The association with lower SES was more pronounced for two components of stigma: 'thinking less of a smoker' and 'not dating a smoker'.

Perceived stigmatization of smokers by others varied a lot between countries, particularly for items related to stereotyping. For example, 81.0% of Dutch adolescents agreed that 'most people think less of a smoker', compared to only 14.2% of Italian adolescents. However, there was no indication that stigmatization was greater in countries with stronger TCP implementation scores. While levels of stigmatization in Ireland (TCP = 70) were similar to those in Germany (TCP = 37), levels of agreement with 'think less of a smoker' were even lower in Ireland (55.9%) than in Germany (73.4%).

Table 2 presents the results of the multilevel linear regression of the perceived stigmatization of smokers by others. Model 1 presents each exposure variable one by one, controlled for gender, age, country, and a random intercept at the school level. Model 2 includes all exposure variables together, with the same controls for confounding.

**Table 1. Agreement with stigmatization components according to adolescents' smoking status and sociodemographic status, SILNE R, 2016, percentage and chi-square test.**

| Covariates | Think less of a smoker | | Think smokers are losers | | Think people would not date a smoker | | Think people would not hire a smoker as a babysitter | | Sample total | |
|---|---|---|---|---|---|---|---|---|---|---|
| | % agreeing | χ2 | % agreeing | χ2 | % agreeing | χ2 | % agreeing | χ2 | % | N |
| **All** | **52.80%** | | **31.40%** | | **56.10%** | | **78.50%** | | | **12712** |
| **Smoking Status** | | | | | | | | | | |
| Non-smokers | | | | | | | | | | |
| Had tried | 56.4 | 102.7** | 34.9 | 141.2** | 61.2 | 255.9** | 82.5 | 308.2** | 65.7 | 8177 |
| Had tried | 49.2 | | 28.3 | | 51.1 | | 74.5 | | 11.8 | 1475 |
| Experimenter | 45.2 | | 24.8 | | 48.4 | | 74.7 | | 5.9 | 735 |
| Ex-smokers | 46.0 | | 26.4 | | 44 | | 75.5 | | 4.9 | 605 |
| Weekly | 46.8 | | 21.7 | | 44.3 | | 67.2 | | 4.4 | 546 |
| Daily | 45.4 | | 21.1 | | 41.9 | | 60.7 | | 7.3 | 912 |
| **Power variables:** | | | | | | | | | | |
| **STP score per school** | | | | | | | | | | |
| Under 5 | 51.5 | 59.4** | 27.5 | 40.0** | 56.8 | 60.3** | 78.3 | 28.2** | 18.9 | 2449 |
| 5 to 7 | 56.7 | | 34.2 | | 59.5 | | 80.5 | | 42.9 | 5577 |
| Over 7 | 49.3 | | 30.3 | | 52.0 | | 76.2 | | 38.2 | 4965 |
| **% of at least weekly smokers in the school** | | | | | | | | | | |
| Under 5% | 59.4 | 258.4** | 44.2 | 750.3** | 67 | 687.1** | 87.6 | 435.8** | 22.7 | 2947 |
| 5% to 10% | 59.9 | | 40.6 | | 66.8 | | 84.1 | | 28.9 | 3755 |
| 10% to 20% | 46.2 | | 19.1 | | 41.8 | | 70.5 | | 32.9 | 4273 |
| Over 20% | 44.0 | | 21.5 | | 50.3 | | 71.1 | | 15.5 | 2016 |
| **Peers who smoked at least weekly** | | | | | | | | | | |
| No friends who smoked | 55.2 | 99.7** | 33.7 | 107.8** | 59.0 | 148.9** | 81.0 | 188.2** | 76.7 | 9961 |
| <50% of friends who smoked | 47.1 | | 25.9 | | 49.0 | | 73.1 | | 13.4 | 1744 |
| > = 50% of friends smoking | 42.4 | | 21.3 | | 43.5 | | 65.8 | | 9.9 | 1286 |
| **Academic performance** | | | | | | | | | | |
| Low | 54.1 | 63.1** | 29.0 | 8.1* | 51.1 | 55.8** | 74.9 | 30.2** | 20.0 | 2548 |
| Medium | 56.5 | | 32.1 | | 54.9 | | 78.3 | | 39.2 | 4987 |
| High | 48.7 | | 31.9 | | 59.7 | | 80.3 | | 40.8 | 5198 |
| **Parents' education** | | | | | | | | | | |
| Low | 45.1 | 168.4** | 27.7 | 51.1** | 48.8 | 147.2** | 74.8 | 63.8** | 34.1 | 4433 |
| Medium | 58.0 | | 32.3 | | 60.4 | | 81.3 | | 43.9 | 5708 |
| High | 54.7 | | 35.5 | | 59.2 | | 78.7 | | 20.5 | 2661 |
| Unknown | 55.8 | | 34.9 | | 53 | | 72.8 | | 1.5 | 189 |
| **Confounders** | | | | | | | | | | |
| **Gender** | | | | | | | | | | |
| Male | 55.9 | 46.9** | 34.7 | 62.4** | 57.1 | 4.7* | 78.5 | 0.0 | 49.7 | 6450 |
| Female | 49.9 | | 28.2 | | 55.2 | | 78.5 | | 50.3 | 6515 |
| **Age group** | | | | | | | | | | |
| 12–14 | 54.1 | 9.4* | 37.3 | 149.7** | 60.8 | 87.3** | 83.9 | 190.3** | 32.6 | 4230 |
| 15 | 51.2 | | 31.6 | | 56.4 | | 79.1 | | 39.9 | 5179 |
| 16+ | 53.7 | | 24.3 | | 50.2 | | 71.0 | | 27.5 | 3564 |
| **Cities** | | | | | | | | | | |
| Dublin (IE) | 55.9 | 2286** | 40.9 | 1480** | 65.9 | 1086** | 84.5 | 524.0** | 16.3 | 2120 |
| Tampere (FI) | 33.7 | | 62.8 | | 75.9 | | 88 | | 13.3 | 1733 |
| Amersfoort (NL) | 81 | | 34.9 | | 65.4 | | 81.1 | | 14.3 | 1858 |

*(Continued)*

**Table 1.** (Continued)

| Covariates | Think less of a smoker | | Think smokers are losers | | Think people would not date a smoker | | Think people would not hire a smoker as a babysitter | | Sample total | |
|---|---|---|---|---|---|---|---|---|---|---|
| | % agreeing | χ2 | % agreeing | χ2 | % agreeing | χ2 | % agreeing | χ2 | % | N |
| **All** | **52.80%** | | **31.40%** | | **56.10%** | | **78.50%** | | | **12712** |
| Latina (IT) | 14.2 | | 21.9 | | 44.8 | | 74.9 | | 15.3 | 1982 |
| Coimbra (PT) | 54.7 | | 14.4 | | 29.7 | | 63.6 | | 14.3 | 1862 |
| Namur (BE) | 60.9 | | 13.4 | | 50.5 | | 71.7 | | 14.9 | 1939 |
| Hanover (DE) | 73.4 | | 34.7 | | 62.8 | | 87.5 | | 11.5 | 1497 |

* P-value < 0.05;

**P-value < 0.001

Countries were ranked according to their implementation of tobacco control policies, the highest scores being those of Ireland (70), Finland (60), the Netherlands (53), Italy (51), and Portugal (50) and the lowest being those of Belgium (49) and Germany (37).

The more experience respondents had of smoking, the less overall stigmatization of smokers they perceived: occasional smokers scored lower on the scale of perceived stigmatization than non-smokers and weekly smokers scored even lower. The school smoking environment had few effects on the perceived stigmatization of smokers by others: stricter STPs were not associated with perceived stigmatization and a higher prevalence of weekly smoking in schools was associated with a slightly lower perceived stigmatization score. The school intra-class

**Table 2.** Stigma and the smoking environment at school among adolescents in seven European cities: Results of multilevel linear regression for the SILNE R study, 2016.

| | Stigmatization (mean score = 1.43 std = 0.38) | |
|---|---|---|
| | Model 1 | Model 2 |
| | β (StdErr) | β (StdErr) |
| **Smoking status** (ref = non-smokers) | | |
| Occasional | -0.071** (0.008) | -0.062** (0.009) |
| Weekly | -0.146** (0.010) | -0.128** (0.012) |
| **Mean STP score per school (0–10)** | 0.005 (0.006) | 0.003 (0.005) |
| **% of weekly smokers in the school** | -0.002* (0.001) | 0.000 (0.001) |
| **Weekly smoking among peers (%)** | -0.141** (0.015) | -0.057** (0.017) |
| **Academic performance** (ref = high) | | |
| Average | -0.038** (0.007) | -0.022* (0.008) |
| Low | -0.070** (0.009) | -0.038** (0.010) |
| **Parents' education** (ref = high) | | |
| Medium | 0.040** (0.009) | 0.038** (0.009) |
| Low | 0.002 (0.009) | 0.011 (0.010) |
| Unknown | 0.003 (0.029) | 0.014 (0.035) |
| **School variance component** | 0.006* (0.000) | 0.007* (0.000) |

For linear regression, we determined an average stigma score by adding together all scores for the four Likert items and dividing by four, using only occasional and weekly smokers in these analyses. In Model 1, using school as a random variable, we controlled for gender, country, and age and each covariate was included once. In Model 2, all covariates are included together.

* P-value < 0.05;

**P-value < 0.001

**Table 3. Stigma and the smoking environment at school among adolescents in seven European cities: Multilevel linear regression stratified by smoking status, SILNE R study 2016.**

| | Stigmatization (mean score = 1.43 std = 0.38) | | |
| --- | --- | --- | --- |
| | Model 2 (non-smokers) | Model 2 (occasional) | Model 2 (weekly) |
| | β (StdErr) | β (StdErr) | β (StdErr) |
| **Mean STP score per school (0–10)** | 0.004 (0.006) | 0.002 (0.010) | 0.010 (0.013) |
| **% of weekly smokers in the school** | 0.001 (0.001) | 0.005* (0.001) | -0.005* (0.002) |
| **Weekly smoking among peers (%)** | -0.082* (0.026) | -0.128** (0.032) | 0.021 (0.037) |
| **Academic performance** (ref = high) | | | |
| Average | -0.021* (0.008) | -0.027 (0.019) | 0.002 (0.030) |
| Low | -0.049** (0.011) | -0.031 (0.022) | 0.015 (0.034) |
| **Parents' education** (ref = high) | | | |
| Medium | 0.034** (0.010) | 0.035 (0.024) | 0.076* (0.037) |
| Low | 0.005 (0.011) | 0.012 (0.024) | 0.061 (0.036) |
| Unknown | 0.028 (0.038) | -0.069 (0.093) | 0.027 (0.133) |
| **School variance component** | 0.007* (0.000) | 0.012 (0.001) | 0.000** (0.007) |

For linear regression, we determined an average stigma score by summing all scores for the four Likert items and dividing by four.

For the stratification, we included only occasional and weekly smokers in these analyses.

* P-value < 0.05;

**P-value < 0.001

correlation, although significant, accounted for less than 1% of the total variance. Being surrounded by a high number of friends who smoked, however, was associated with less perceived stigmatization of smokers by others (Beta = -0.141, $p < 0.001$). The poorer the adolescents' academic performance was, the lower was the perceived stigmatization of smokers they reported. However, this gradient was not present in relation to parental education: the students with the greatest perceived stigmatization of smokers were those whose parents had received a medium level of education.

The results for Model 2 were similar to those for Model 1, with somewhat lower coefficients in absolute value. Less stigmatization of smokers was perceived by students who smoked, those whose academic performance was low, and those who had friends who smoked. While stigmatization decreased for those who had friends who smoked, it was not associated with weekly smoking at school.

As testing showed that smoking status and a school's SES interacted on overall perceived stigmatization of smokers (F-test = 7.9, $p = 0.0003$), we stratified the previous analysis by smoking status (Table 3) and school SES (Table 4). This showed that less perceived stigmatization of smokers was reported by occasional smokers with friends who smoked (Table 3). Table 4 shows that weekly smokers perceived the stigmatization of smokers less in low-SES schools than in high-SES schools (-0.160 *vs* -0.095), thereby supporting our hypothesis that, as smoking becomes increasingly denormalized, smokers may be vulnerable to stigmatization. In high-SES schools, having friends who smoked was a protective factor against stigma.

## Discussion

### Main findings

In a unique international survey involving seven European cities, we examined how perceived stigmatization of smokers among adolescents was affected by their smoking environment and the power imbalance between smokers and non-smokers. We found conclusive evidence that

**Table 4. Stigma and the smoking environment at school among adolescents in seven European cities: Multilevel linear regression stratified by school SES status, SILNE R study 2016.**

| Stigmatization | Low-SES schools Model 2 | High-SES schools Model 2 |
|---|---|---|
| | β (StdErr) | β (StdErr) |
| **Smoking status** (ref = non-smokers) | | |
| Occasional | -0.034* (0.016) | -0.075** (0.010) |
| Weekly | -0.160** (0.019) | -0.095** (0.015) |
| **Mean STP score per school (0–10)** | 0.007 (0.012) | 0.002 (0.007) |
| **% of weekly smokers in the school** | -0.003 (0.002) | -0.001 (0.002) |
| **Weekly smoking among peers (%)** | -0.046 (0.026) | -0.066* (0.023) |
| **Academic performance** (ref = high) | | |
| Average | -0.038* (0.014) | -0.017 (0.009) |
| Low | -0.052* (0.017) | -0.034* (0.012) |
| **Parents' education** (ref = high) | | |
| Medium | 0.052* (0.017) | 0.034* (0.011) |
| Low | -0.009 (0.015) | 0.023 (0.012) |
| Unknown | 0.043 (0.055) | -0.020 (0.046) |
| **School variance component** | 0.007 (0.001) | 0.003 (0.000) |

For the linear regression, we determined an average stigma score by adding together all scores for the four Likert items and dividing by four, using only occasional and weekly smokers in these analyses.

* P-value < 0.05;

**P-value < 0.001

smokers in the adolescent population at large were aware of the negative effects of stigmatization, stereotyping, and discrimination. Students who smoked reported less overall perceived stigmatization of smokers than non-smokers did. Perceived stigmatization of smokers was lower among those who were more exposed to smoking among their friends and among those with poorer academic performance. We found no indication that stronger STPs were associated with more stigmatization.

## Consistency and interpretation

Our study found that weekly smokers expressed the lowest perceived stigmatization of smoking and that non-smokers expressed the highest. One possible explanation is that adolescents for whom smoking is a key component of their identity may actively resist stigmatization by denying their identity as a smoker. A growing number of experts who warn public health authorities against the use of stigmatization as a preventative tool have highlighted two negative consequences of stigmatization: social isolation from non-smokers and concealment of smoking status [39, 40]. In a qualitative study of adults who engaged in 'social smoking,' Whitesel and Shuman showed that smoking in groups allowed individuals to identify as non-smokers even while they continued to smoke [41]. To hide their smoking status, individuals who were unwilling to be categorized as smokers used neutralization techniques that had previously been identified by Becker [41]. For example, adolescents may use neutralization techniques, such as asking others to buy cigarettes for them or obtaining cigarettes from social sources and sharing the packet, to reduce the effects of denormalization and the fear of stigma.

 Occasional smokers tend not to smoke alone and tend to use social motivations as a rationale for smoking ('to be with friends'), largely to feel more in control and to minimize the risk both of becoming dependent [42] and of social isolation, a consequence of stigmatization. Our

results showed that, whereas weekly smokers were not protected against stigmatization by having peers who smoked, occasional smokers whose peers were weekly smokers did indeed perceive significantly less stigmatization. We also found that discrimination was perceived less by smokers than by non-smokers. Perceived discrimination against smokers by others ('not hiring a smoker as babysitter') was more common than perceived stereotyping of smokers ('most people think less of people who smoke').

Our findings indicate that less stigmatization was felt by adolescents who smoked, adolescents whose academic performance was poor, and those who had friends who smoked. This might be consistent with the concept of 'smoking islands', i.e. cohesive socioeconomically disadvantaged groups in which smoking is more accepted, either due to a sense of active resistance or because people in these groups feel powerless to quit smoking [21]. Our findings suggest that adolescents who smoked felt the effects of stigmatization less in low-SES schools than in high-SES schools, regardless of the impact of STPs. Again, this suggests that social context contributes to stigmatization. It is consistent with the conclusion of Stuber's study, which found that smokers who live in segregated neighbourhoods perceive less stigmatization [32]. As indicated by Farrimond, smoking remained more accepted in low-SES areas, where rates of smoking were higher. Thus, 'felt stigmatization' was weaker among low-SES smokers than among high-SES smokers [43].

Adolescents with the highest individual academic performance or parental SES expressed greater agreement with stigmatization of smoking than those with a lower SES. The greater sensitivity to perceived stigmatization on the part of smokers from high and medium SES backgrounds enables us to understand the differential impact of the 'social unacceptability' of smoking within this population. These findings indicate that stigmatization is a mechanism whereby underlying 'social unacceptability' may contribute to disparities through 'denormalization' strategies [21]. If smokers who are socioeconomically disadvantaged feel less stigmatization, the intended mechanism of social control will not operate as effectively for that population. This may explain why smoking rates in disadvantaged communities are not currently decreasing any further, despite growing stigma [44]. Aside from the school context and the peer context, it could also be explained by the role of the tobacco industry, which has been targeting young smokers of lower SES [45–48]. Current smokers could thus be subjected to a double stigma: smoking and low SES [49]. If so, 'denormalization' strategies may eventually exacerbate health inequalities and further marginalize smokers.

Finally, the hypothesis that STPs would be associated with greater perceived stigmatization of smokers by others was not confirmed. This might contradict the literature on the role of political power (STPs) in the stigmatization process. Stigmatization results from the enforcement by social institutions (such as schools) of the use of stigmatization as a preventative tool and from the resulting increase in the social distance between smokers and non-smokers [50]. One possible explanation is that STPs limit the experience of stigmatization to certain aspects of daily life and to certain situations and may ultimately cause only minor inconvenience. Indeed, the stigmatization associated with smoking is reversible: for stigmatization to cease, it is enough for an individual to leave the school premises. STPs were not found to decrease smoking just outside school premises, meaning that smoking remained visible [33]. Moreover, exposure to friends smoking outside school premises limits the effect of denormalization on smoking.

## Policy implications

Although denormalization could be expected to increase stigmatization of smokers, our study showed that stigmatization was not increased by School Tobacco Policies. This means that

current school tobacco policies, which mainly focus on smoking bans and sometimes on health education, do not create stigmatization. School smoking prevention programmes that make use of social competences to counter smoking-positive stereotypes (smoking is cool, smoking is for rebels, etc.) [51] and focus on peer influence as a way of reducing smoking initiation [52, 53] could, however, be a useful addition to current tobacco prevention policies in all schools, not only in low-SES schools [54].

Further research could examine whether smoking prevention strategies and smoking cessation services for staff and students have an impact on the stigmatization of smokers [55]. It would also be useful to study whether users of e-cigarettes and other new nicotine delivery systems encounter stigmatization similar to that experienced by smokers. Similarly, comparison with users of smokeless tobacco, such as Swedish snus, may provide interesting findings because their use is invisible or is less easily observed by others.

## Strengths and limitations

There are several ways in which this paper adds to the previous body of knowledge. First, it draws on a large international survey carried out in 55 schools in seven countries, each of whose tobacco control policies differs with regard to the strictness of its implementation. The design of the study allowed us to capture the role of power in the association between stigma and smoking. This is one of the first studies to apply a validated scale of stigmatization to adolescents (smokers and non-smokers).

Our scale of stigma measures what people believe to be the social norm, not their personal opinions. As Stuber's scale was designed for adults, its wording was revised in order to make it suitable for surveying adolescents in different countries. This methodological issue could help to explain the large differences between countries. The adaptation of Stuber's scale and its translation into the languages of the seven European countries may have led adolescents to understand terms such as 'losers' in different ways. It could also be argued that Stuber's scale does not measure real experience of discrimination so much as rational responses to the harmful effects of passive smoking on innocent third parties, such as children (Stuber, Galea, et al., 2008). Given these limitations, the score serves mainly as a general proxy of public stigmatization.

## Conclusion

Perceived stigmatization of smoking is common among European adolescents. Smokers themselves, however, perceive stigma less often than non-smokers. Denormalization of smoking in the school environment does not increase stigmatization or protect against it but being in a social environment that is permissive of smoking decreases the perception of stigmatization. Our findings suggest that it may be possible to create school tobacco policies that change the smoking climate without contributing to the stigmatization of individual smokers. It is also interesting to note that social capital mitigates the effects of denormalization strategies and protects smokers from stigmatization.

## Supporting information

**S1 Table. Participation rates among students by city, SILNE-R study in seven EU cities, 2016.**
(DOCX)

**S2 Table. STP dimensions construction, SILNE-R study in seven EU cities.**
(DOCX)

**S1 File. Ethical approvals.**
(DOCX)

**S2 File. The original written English adolescent questionnaire.**
(PDF)

## Acknowledgments

We would like to acknowledge the colleagues involved not only in the collection of the SILNE WP5 data—Jaana Kinnunen, Geatano Roscillo, Pierre-Olivier Robert, and Joanna Alves—but also of the SILNE-R WP8 data: Pauline A.W. Nuyts, Michael Schreuders, Diego Marandola, Teresa Leão, Pierre-Olivier Robert, Adeline Grard, Jaana Kinnunen, Laura Hoffmann, and Martin Mlinarić. Our acknowledgements for the data management are also due to Mostafa Berdii and Adeline Grard. As well as thanking Séverine Guisset and the SMCS-UC Louvain team for their precious assistance with statistics, we would like to thank all the school staff and students who participated in the study.

## Author Contributions

**Conceptualization:** Adeline Grard, Anton E. Kunst.

**Data curation:** Martin Mlinarić.

**Formal analysis:** Adeline Grard, Nora Mélard.

**Project administration:** Anton E. Kunst.

**Supervision:** Anton E. Kunst, Vincent Lorant.

**Validation:** Arja Rimpelä, Matthias Richter.

**Writing – original draft:** Pierre-Olivier Robert.

**Writing – review & editing:** Nora Mélard, Arja Rimpelä, Matthias Richter, Vincent Lorant.

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
