## [Decision Letter · Decision Letter 0]

17 Mar 2020

PONE-D-20-02662

The effect of school smoke-free policies on smoking stigmatization: a European comparison study among adolescents.

PLOS ONE

Dear Mr. Robert,

Thank you for submitting your manuscript to PLOS ONE. After careful consideration, we feel that it has merit but does not fully meet PLOS ONE’s publication criteria as it currently stands. Therefore, we invite you to submit a revised version of the manuscript that addresses the points raised during the review process.

In addition to addressing the specific technical points that the reviewers raised, please discuss the policy implications for your findings.  Denormalization of tobacco use (and the tobacco industry) and social norm change has been found to be an effective way to reduce smoking among youth (and adults).  What are the implications of the results in this paper for implementing such strategies in the best way to minimize advser side effects?

We would appreciate receiving your revised manuscript by May 01 2020 11:59PM. To enhance the reproducibility of your results, we recommend that if applicable you deposit your laboratory protocols in protocols.io, where a protocol can be assigned its own identifier (DOI) such that it can be cited independently in the future. For instructions see: http://journals.plos.org/plosone/s/submission-guidelines#loc-laboratory-protocols

We look forward to receiving your revised manuscript.

Kind regards,

Stanton A. Glantz

Academic Editor

PLOS ONE

Journal Requirements:

2. Please correct your reference to "p=0.000" to "p<0.001" or as similarly appropriate, as p values cannot equal zero.

3. In your Methods section, please provide additional information in the "confounders" section of your Methods, explaining in more detail how different countries received their scores.

Reviewers' comments:

Reviewer's Responses to Questions

**Comments to the Author**

1. Is the manuscript technically sound, and do the data support the conclusions?

Reviewer #1: Yes

Reviewer #2: Yes

2. Has the statistical analysis been performed appropriately and rigorously? 

Reviewer #1: Yes

Reviewer #2: Yes

3. Have the authors made all data underlying the findings in their manuscript fully available?

Reviewer #1: Yes

Reviewer #2: No

4. Is the manuscript presented in an intelligible fashion and written in standard English?

Reviewer #1: Yes

Reviewer #2: Yes

5. Review Comments to the Author

Reviewer #1: The authors address an interesting issue, and test their hypotheses with a unique and international data system. I think that the paper could be improved with minor revisions to include more detail.

1. Methods, first paragraph

- Did every school in each city participate? if not, how were schools sampled?

- Was very student in each school eligible to participate?

-"The response rate in the survey was 80%" Can you provide more detail? Did it vary widely across cities?

2. Methods, page 9

- Regarding the stigmatization index, why were the four response options dichotomized prior to creating a summed score? A outcome score that ranged from 0 to 12 would be more useful than one that ranged from 0 to 4

- or is this average and not a sum score

- The text states that this stigma score was created by summing the 4 dichotomized variables (but does not specify that this score was then divided by 4). It also refers to this variable as an average. Is it a sum or average? What is its range?

- Why was this stigma score than log-transferred?

3. Methods, page 9

- Which school staff provided perceptions for the STP? How were they selected? Did their STPs correlate well with those of the students in their school? Did students within the same school have similar STP responses?

4. Methods, page 10

- Can you provide more detail about the calculation of the TCP scores?

Reviewer #2: This is an interesting, albeit long, manuscript describing the potential stigmatization of European adolescent smokers in their school environment. It does so by embedding it in sociological theory.

Comments are in chronological order.

Background in the abstract: it is stated that it is unclear whether there is stigmatization. Then, the first aim of the paper is to measure if stigmatization varies according to smoking status etc… This assumes that there actually is proven stigmatization. So, this does not follow from what you write in the sentence before.

I find it odd that the results section of the abstract does not include any numbers.

First paragraph of the introduction section: I think, it would be more accurate to change “public health policies” to “tobacco control policies”. And, while it is true that such policies have been developed and supported by public health organizations, one example being the FCTC, it really is much more about what gets implemented in individual countries on the country-level and on more local level.

Second paragraph of the introduction: a comprehensive package of tobacco control policies certainly also includes taxation and other things, so this term is much broader than what the authors define as social denormalization strategies.

Towards the end of the Introduction, and a general comment for the entire manuscript: the authors discuss the stigma and potential disparities/inequalities among smokers associated with the denormalization of smoking through tobacco control measures. However, they do not discuss the role of the tobacco industry and their decade-long and ongoing influence and how it might have influenced many of the believes the authors are analyzing. The industry has done a phenomenal job targeting the populations discussed here, youth and lower SES.

Also, this is about cigarette smoking only. What about non-cigarette products that are popular among youth? And what about their current normalization through young social media influencers and other channels?

It seems like parts of the Introduction already have sections on Theoretical Background, which is its own section. This should be streamlined and shortened. The Theoretical Background section should generally be shortened and more to the point, rather than just describing previous research.

Methods section: did parents have to give consent for their children to participate?

First paragraph of the Data Analysis section: you might want to give a reference for multilevel analysis or explain briefly, so the readers who are less familiar with multilevel analysis will know what it e.g. means when you use “school” as a random intercept.

Towards the end of this section: it is really the students in a school that might have higher or lower SES on average. A school cannot have a socioeconomic status. Or is this maybe not what you mean?

Discussion/main findings section: I am not sure I understand what you are trying to imply by using the term “double-edge sword”. Please be more clear.

Consistency and interpretation section: I would be interested to see more literature discussed about the potential association of stigmatization and quitting smoking among adolescents. Also, please be more to the point. For example, what are those neutralization techniques you mention and what does it possibly imply for your study and adolescent smoking? Why are those techniques relevant here?

End of first paragraph on page 22: it sounds like you are implying that social stigma is the only mechanism driving down smoking rates. How about reaching those lower SES populations with higher smoking prevalence with appropriate interventions, be it in terms of education, or in terms of access to cessation aids, among other things?

6. PLOS authors have the option to publish the peer review history of their article (what does this mean?). If published, this will include your full peer review and any attached files.

Reviewer #1: No

Reviewer #2: No

---

## [Author Response · Author response to Decision Letter 0]

12 Jun 2020

Comment 1: In addition to addressing the specific technical points that the reviewers raised, please discuss the policy implications for your findings. Denormalization of tobacco use (and the tobacco industry) and social norm change has been found to be an effective way to reduce smoking among youth (and adults). What are the implications of the results in this paper for implementing such strategies in the best way to minimize adverse side effects?

Reply : We agree. We have added a section about policy implications to the discussion section of our manuscript. This now reads:

“Although denormalization could be expected to increase stigmatization of smokers, our study showed that stigmatization was not increased by School Tobacco Policies. This means that current school tobacco policies, which mainly focus on smoking bans and sometimes on health education, are safe in terms of stigmatization. School smoking prevention programmes that make use of social competences to counter smoking-positive stereotypes (smoking is cool, smoking is for rebels, etc.) (1) and focus on peer influence as a way of reducing smoking initiation (2, 3) could, however, be a useful addition to tobacco prevention policies in all schools, not only in low SES schools (4).

Further research could examine whether smoking prevention strategies and smoking cessation services for staff and students have an impact on the stigmatization of smokers (5). It would also be useful to study whether users of e-cigarettes and other new nicotine delivery systems encounter stigmatization similar to that experienced by smokers. Similarly, comparison with users of smokeless tobacco, such as Swedish snus, could provide interesting findings because their use is invisible or is less easily observed by others.”

Comment 2: To enhance the reproducibility of your results, we recommend that if applicable you deposit your laboratory protocols in protocols.io, where a protocol can be assigned its own identifier (DOI) such that it can be cited independently in the future. For instructions see: http://journals.plos.org/plosone/s/submission-guidelines#loc-laboratory-protocols

Reply: The protocol has been already explained elsewhere: Lorant V, Soto VE, Alves J, Federico B, Kinnunen J, Kuipers M, et al. Smoking in school-age adolescents: Design of a social network survey in six European countries. BMC Research Notes. 2015;8 (1).

 

Journal additional Requirements:

Comment 1: Please ensure that your manuscript meets PLOS ONE's style requirements, including those for file naming. The PLOS ONE style templates can be found at http://www.journals.plos.org/plosone/s/file?id=wjVg/PLOSOne_formatting_sample_main_body.pdf and http://www.journals.plos.org/plosone/s/file?id=ba62/PLOSOne_formatting_sample_title_authors_affiliations.pdf

Please include captions for your Supporting Information files at the end of your manuscript, and update any in-text citations to match accordingly. Please see our Supporting Information guidelines for more information: http://journals.plos.org/plosone/s/supporting-information.

Reply: The abstract, captions, supporting information files, and paper sections have been formatted according to the journal’s editorial requirements.

Comment 2: Please correct your reference to "p=0.000" to "p<0.001" or as similarly appropriate, as p values cannot equal zero.

Reply: We have now modified our tables accordingly. 

Comment 3: In your Methods section, please provide additional information in the "confounders" section of your Methods, explaining in more detail how different countries received their scores.

Reply: We have now added this information in the confounders section: 

“Countries were ranked according to the Tobacco Control Scale (TCS) developed by Joossens and Raw, ranging from high scores (Ireland-70 and Finland-60) to medium (the Netherlands-53, Italy-51, and Portugal-50) and low scores (Belgium-49 and Germany-37). The TCS includes factors such as taxes, restrictions on smoking in public places, consumer information (e.g. public information campaigns, media coverage, publicizing research findings), comprehensive bans on the advertising of all tobacco products, labels with health warnings on tobacco products, and treatment to help dependent smokers to quit. The scale allocates points to each policy, with a maximum score of 100: prices 30, smoke-free public places 22, spending on public information campaigns 15, comprehensive advertising bans 13, large health warnings 10, cessation support (treatment) 10 (Joossens and Raw, 2017). However, the TCS score does not take school smoking bans into account. That is why we created a school tobacco policy score (STP) for each school (6).” We have provided, in S2 Table, a detailed list of all items of this STP score (6). 

Comment 4: We note that you have indicated that data from this study are available upon request. PLOS only allows data to be available upon request if there are legal or ethical restrictions on sharing data publicly. For information on unacceptable data access restrictions, please see http://journals.plos.org/plosone/s/data-availability#loc-unacceptable-data-access-restrictions. 

Reply: There are ethical restrictions on sharing data which contain potentially personally identifiable information pertaining to adolescents. We have followed your advice and we have now added the list of ethical approvals in the S1 File. Information about the ethical and legal restrictions has been included in the manuscript. We refer to this in the methods section: 

“In each city, ethical approval was obtained from (sub-) national or local organizations, for which full references are provided (See S1 File). In some cities, permission to conduct the survey was also requested from the educational authorities. In accordance with each country’s regulations, school principals, parents, and adolescents received leaflets, information letters, and parental consent letters. All study participants were asked to sign an informed consent agreement prior to participating in the study.” 

All study participants were asked to sign an informed consent agreement prior to participating in the study. Following the recommendations of the European Commission, national and local organizations, the datasets generated in this study are not publicly available. Making all the data available would contravene the Consortium Agreement that was developed and signed before the start of the SILNE-R project, specifically with regards to the ownership and use of the project output. Furthermore, we would like to retain control over the use of the data in order to avoid it being used inappropriately by the tobacco industry. In 2021, however, any person can apply for access to the data through the SILNE R consortium (the data owner). The contact person, as data holder, is Anton E Kunst, a.e.kunst@amsterdamumc.nl. Within the Institute, the two contact persons are: Regina Below, from the research department, regina.below@uclouvain.be; and Alaa Mahboub, for technical support, alaa.mahboub@uclouvain.be.

Reviewers' comments:

Reviewer #1: The authors address an interesting issue, and test their hypotheses with a unique and international data system. I think that the paper could be improved with minor revisions to include more detail.

Thank you very much for these valuable and encouraging comments. Some important points have been raised. We have done our best to properly address all your concerns below.

Comment 1: Did every school in each city participate? if not, how were schools sampled? ….Was every student in each school eligible to participate?..."The response rate in the survey was 80%" Can you provide more detail? Did it vary widely across cities?

Reply: Thank you for pointing this out. We have now added this information in the methods section.

“This research was based on the ‘SILNE-R’ study, which was conducted in 2016/2017 in seven European cities. In each city, schools were selected from the local register of schools, and were categorized as either high- or low- SES schools. Six to twelve schools were selected and participated in the SILNE R survey. The schools were stratified according to information that was specific to each country: either the type of school (Italy, Germany, and the Netherlands) with (non-)academic tracks, the socio-economic ranking of the school by the educational authorities (Belgium and Portugal), or the area’s socio-economic status (Finland). Data were collected in 55 schools from 12,991 adolescents. In each school, two groups of school students were selected from grades corresponding to the ages of 14 to 16. In those two grades, all registered adolescents were invited to participate in the survey.”

There was a lack of information about the difference in response rates across cities in the article. In 2016, participation rates among adolescents were over 84% in Namur – Belgium, Amersfoort – the Netherlands, and Tampere, Finland, and were 80% in Dublin – Ireland, 79% in Latina – Italy, 76% in Coimbra – Portugal, and 66% in Hanover – Germany. In Germany and Italy, active parental consent was required, which resulted in a lower response rate compared to that in the cities where passive consent sufficed. Adolescents were considered non-participants if they were absent on the day of the study or unwilling to participate and were excluded if their questionnaires were blank or seemed too incoherent. Following your comment 1, we decided to include the following table as a supplementary table. It is a summary of the participation rates among adolescents in each city. We now refer to this in the methods section: “See S1 Table for more information on the response rates.”

S1 Table Participation rates among students by city, SILNE-R study in seven EU cities, 2016

 Cities – Countries 2016

 N Participation rate (%) Std

(%) Min

(%) Max

(%)

Students sample 

Namur – Belgium (Nschools = 7) 1939 84.1 0.76 82.5 85.5

Tampere – Finland (Nschools = 9) 1733 87.1 0.75 85.5 88.5

Hanover – Germany (Nschools = 12) 1497 65.8 0.99 63.8 67.7

Latina – Italy (Nschools = 7) 1982 78.9 0.81 77.3 80.5

Amersfoort – The Netherlands (Nschools = 6) 1858 84.9 0.76 83.4 86.4

Coimbra – Portugal (Nschools = 6) 1862 76.2 0.86 74.5 77.8

Ireland – Dublin (Nschools = 8) 2120 80.3 0.77 78.7 81.7

Comment 2: Regarding the stigmatization index, why were the four response options dichotomized prior to creating a summed score? A outcome score that ranged from 0 to 12 would be more useful than one that ranged from 0 to 4….or is this average and not a sum score…The text states that this stigma score was created by summing the 4 dichotomized variables (but does not specify that this score was then divided by 4). It also refers to this variable as an average. Is it a sum or average? What is its range? Why was this stigma score than log-transferred? 

Thank you for this comment. We now explain more precisely how the score was computed. We have also provided more details regarding the statistical analysis. The four replies (from strongly agree to strongly disagree) were rated from 0 to 3. Then, an average stigma score was determined by adding together all the scores for the four Likert items and dividing by four (mean = 1.62, std = 0.55). The score was then log-transformed to reduce skewness and facilitate parametric tests such as linear regression. 

Comment 3: Which school staff provided perceptions for the STP? How were they selected? Did their STPs correlate well with those of the students in their school? Did students within the same school have similar STP responses?

Reply: Our study protocol stated that we would administer the questionnaire to at least three staff members per school, including, where possible, someone from the management, someone from the teaching staff, and someone in charge of health and prevention matters. In this study, between 24 and 44 school staff members participated in each city.

Overall, staff and student perceptions of STPs were significantly correlated (r = 0.46, p = 0.0034). Their scores were significantly correlated for policy comprehensiveness too (r = 0.55, p = 0.0003) and for policy enforcement (r = 0.51, p = 0.0010). We have now added this information in the methods section.

This aspect has been addressed in detail in another forthcoming paper (accepted for publication in Preventive Medicine) : “School tobacco policies and adolescent smoking in six European cities in 2013 and 2016: A school-level longitudinal study” (6). 

Comment 4: Can you provide more detail about the calculation of the TCP scores?

Reply: See the section quoted above in response to “Journal Additional Requirements”, comment 3. 

Reviewer #2: This is an interesting, albeit long, manuscript describing the potential stigmatization of European adolescent smokers in their school environment. It does so by embedding it in sociological theory.

Comment 1: Background in the abstract: it is stated that it is unclear whether there is stigmatization. Then, the first aim of the paper is to measure if stigmatization varies according to smoking status etc… This assumes that there actually is proven stigmatization. So, this does not follow from what you write in the sentence before. I find it odd that the results section of the abstract does not include any numbers.

Reply: The context of our paper is the increasing denormalization of smoking, not stigmatization. It could be that decades of tobacco denormalization polices have contributed to creating stigmatization. There are several qualitative studies that suggest this link. There are, however, very few quantitative studies that effectively test this association and none of those have focused on adolescents. 

The abstract sections have been amended in line with your suggestions.

 Comment 2: First paragraph of the introduction section: I think, it would be more accurate to change “public health policies” to “tobacco control policies”. And, while it is true that such policies have been developed and supported by public health organizations, one example being the FCTC, it really is much more about what gets implemented in individual countries on the country-level and on more local level.

Reply: Thank you for raising this issue, as this is a core element of the whole SILNE-R research project. In light of your comments on the introduction, we have now decided to shorten the introduction and cut the first and second paragraphs in order to focus more on the literature about stigmatization. Moreover, we changed “public health policies” consistently to “tobacco control policies”. We agree that the implementation of TCPs at a local level is as important as the policy itself. That is why we measured the implementation of STPs and used the score that we developed to evaluate the strength of the tobacco policy in each school in our sample.

Comment 3: Second paragraph of the introduction: a comprehensive package of tobacco control policies certainly also includes taxation and other things, so this term is much broader than what the authors define as social denormalization strategies.

Reply: Thank you for this observation. In light of your comments on the introduction, we have now decided to shorten the introduction and cut the first and second paragraphs in order to focus more on the literature about stigmatization. We have also changed our first paragraph in line with your suggestions. This now reads:

“Denormalization strategies, designed to influence social norms in order to promote reduction of smoking in society (7), have been developed and supported by numerous public health organizations, such as the Centers for Disease Control (CDC) and the World Health Organization (WHO) (8). These strategies restrict smoking in public places, limit access to tobacco products through the prohibition of sales, and set regulations for tobacco advertising. They can also include media campaigns that raise awareness of the dangers of passive smoking (9). One of the underlying assumptions on which they operate is that smokers who experience the stigma of smoking are more likely to quit than those who do not (10). As a consequence of the denormalization of smoking, the general public has developed a negative attitude towards smoking (9, 11, 12).”

Comment 4: Towards the end of the Introduction, and a general comment for the entire manuscript: the authors discuss the stigma and potential disparities/inequalities among smokers associated with the denormalization of smoking through tobacco control measures. However, they do not discuss the role of the tobacco industry and their decade-long and ongoing influence and how it might have influenced many of the believes the authors are analyzing. The industry has done a phenomenal job targeting the populations discussed here, youth and lower SES.

Reply: Thank you for highlighting this important issue. We agree that, encouraged by the industry, smoking was associated with positive values such as pleasure, sociability, and conviviality among young people. This is not to deny the key and detrimental role of the industry, which has been targeting young smokers of lower SES populations. We now refer to this in the discussion section:

“Aside from the school context and the peer context, it could also be explained by the role of the tobacco industry, which has been targeting young smokers of lower SES (13-16).”

Comment 5: Also, this is about cigarette smoking only. What about non-cigarette products that are popular among youth? And what about their current normalization through young social media influencers and other channels?

Reply: Thank you for raising this question. We agree that non-cigarette products such as Alternative Nicotine Delivery Systems (ANDS) are becoming an important concern today. Unfortunately, the questions measuring stigma were not explicit about e-cigarette use and ANDS. We have added a sentence about new nicotine delivery products in the part of the Policy implications section concerning future studies.

Comment 6: It seems like parts of the Introduction already have sections on Theoretical Background, which is its own section. This should be streamlined and shortened. The Theoretical Background section should generally be shortened and more to the point, rather than just describing previous research.

Reply: See our reply to comment 2. Thank you for your remark. The introduction has been streamlined and reduced accordingly. We believe that our introduction has been shortened significantly without affecting the theoretical framework.

Comment 7: Methods section: did parents have to give consent for their children to participate?

Reply: In some countries, permission to conduct the survey was requested from educational authorities. School principals, parents, and adolescents received leaflets, information letters, and parental consent letters, according to each country’s regulations. Active parental consent was required in Italy and Germany. The lower response rate in Germany was partly due to the requirement for active parental consent: parents had to fill in a form that stated whether their children were allowed to participate. We now refer to this in the methods section: 

“In each city, ethical approval was obtained from (sub-) national or local organizations, for which full references are provided (See S1 File). In some cities, permission to conduct the survey was also requested from the educational authorities. In accordance with each country’s regulations, school principals, parents, and adolescents received leaflets, information letters, and parental consent letters. All study participants were asked to sign an informed consent agreement prior to participating in the study.”

Comment 8: First paragraph of the Data Analysis section: you might want to give a reference for multilevel analysis or explain briefly, so the readers who are less familiar with multilevel analysis will know what it e.g. means when you use “school” as a random intercept.

Reply: Multilevel models can be used when observations are not independent, for instance due to a hierarchical structure in the data. A random intercept or random effects may be included in a model in order to take into account the correlation between observations for a single individual, school, or hospital.

We now refer to this in the methods section and we have included this reference: Brown, H. and Prescott, R. (1999) Applied Mixed Models in Medicine, Wiley.

Comment 9: Towards the end of this section: it is really the students in a school that might have higher or lower SES on average. A school cannot have a socioeconomic status. Or is this maybe not what you mean?

Reply: We have now made this more explicit in the methods section: “In each city, schools were selected from the local register of schools, and were categorized as either high- or low- SES schools. Six to twelve schools in each city were selected and participated in the SILNE R survey. The schools were stratified according to information that was specific to each country: either the type of school (Italy, Germany, and the Netherlands) with (non-)academic tracks, the socioeconomic ranking of the school by the educational authorities (Belgium and Portugal), or the area’s socio-economic status (Ireland and Finland).”

Comment 10: Discussion/main findings section: I am not sure I understand what you are trying to imply by using the term “double-edge sword”. Please be more clear.

Reply: We have removed those words. The sentence now reads: 

“We found conclusive evidence that smokers in the adolescent population at large were aware of the negative effects of stigmatization, stereotyping, and discrimination.”

Comment 11: Consistency and interpretation section: I would be interested to see more literature discussed about the potential association of stigmatization and quitting smoking among adolescents. Also, please be more to the point. For example, what are those neutralization techniques you mention and what does it possibly imply for your study and adolescent smoking? Why are those techniques relevant here?

Reply: 

We decided not to discuss the potential association of stigmatization and quitting smoking among adolescents. Stigma may be “functional” in tobacco control, in the sense that the fear of stigma may motivate smokers to quit and deter young people from smoking initiation (17). Among the motives for quitting reported by smokers are the negative image that smoking has in society, a motive more often expressed by more highly educated smokers (18). However, the causal relationship between intention to quit and perceived stigma is not known, nor can it be adequately addressed in our study because of the cross-sectional perspective and the low number of adolescents who want to quit. 

Thank you for this comment, which has given us an opportunity to explain the neutralization technique more precisely: 

“For example, adolescents may use neutralization techniques, such as asking others to buy cigarettes for them or obtaining cigarettes from social sources and sharing the packet, to reduce the effects of denormalization and the fear of stigma.”

Comment 12: End of first paragraph on page 22: it sounds like you are implying that social stigma is the only mechanism driving down smoking rates. How about reaching those lower SES populations with higher smoking prevalence with appropriate interventions, be it in terms of education, or in terms of access to cessation aids, among other things?

Reply: We hope we have addressed this in the changes we described to the policy implications section of our findings. 

“Although denormalization could be expected to increase stigmatization of smokers, our study showed that stigmatization was not increased by School Tobacco Policies. This means that current school tobacco policies, which mainly focus on smoking bans and sometimes on health education, do not create stigmatization. School smoking prevention programmes that make use of social competences to counter smoking-positive stereotypes (smoking is cool, smoking is for rebels, etc.) (1) and focus on peer influence as a way of reducing smoking initiation (2, 3) may, however, be a useful addition to current tobacco prevention policies in all schools, not only in low SES schools(4).

“Further research could examine whether smoking prevention strategies and smoking cessation services for staff and students have an impact on the stigmatization of smokers (5). It would also be useful to study whether users of e-cigarettes and other new nicotine delivery systems encounter similar stigmatization similar to that experienced by smokers. Similarly, comparison with users of smokeless tobacco, such as Swedish snus, may provide interesting findings because their use is invisible or is less easily observed by others.”

References

1. Thomas RE, McLennan J, Perera R. Cochrane in context: School-based programmes for preventing smoking. Evidence-Based Child Health. 2013;8(5):2041-3.

2. Carson KV, Brinn MP, Labiszewski NA, Esterman AJ, Chang AB, Smith BJ. Community interventions for preventing smoking in young people. Cochrane Database of Systematic Reviews. 2011;2017(12).

3. Thomas R, Perera R. School-based programmes for preventing smoking. Cochrane database of systematic reviews (Online). 2006;3.

4. Mercken L, Moore L, Crone MR, De Vries H, De Bourdeaudhuij I, Lien N, et al. The effectiveness of school-based smoking prevention interventions among low- and high-SES European teenagers. Health Education Research. 2012;27(3):459-69.

5. Hefler M, Liberato SC, Thomas DP. Incentives for preventing smoking in children and adolescents. Cochrane Database of Systematic Reviews. 2017;2017(6).

6. Mélard N, Grard A, Robert P-O, Kuipers MAG, Schreuders M, Rimpelä AH, et al. School tobacco policies and adolescent smoking in six European cities in 2013 and 2016: A school-level longitudinal study. Preventive Medicine. 2020:106142.

7. Sæbø G. Tobacco denormalisation and representations of different tobacco users in Norway: A cross-sectional study. Sociology of Health and Illness. 2016;38(3):360-79.

8. Evans-Polce RJ, Castaldelli-Maia JM, Schomerus G, Evans-Lacko SE. The downside of tobacco control? Smoking and self-stigma: A systematic review. Social Science and Medicine. 2015;145:26-34.

9. Bell K, Salmon A, Bowers M, Bell J, McCullough L. Smoking, stigma and tobacco 'denormalization': Further reflections on the use of stigma as a public health tool. A commentary on Social Science & Medicine's Stigma, Prejudice, Discrimination and Health Special Issue (67: 3). Social Science and Medicine. 2010;70(6):795-9.

10. O'Connor RJ, Rees VW, Rivard C, Hatsukami DK, Cummings KM. Internalized smoking stigma in relation to quit intentions, quit attempts, and current e-cigarette use. Substance Abuse. 2017;38(3):330-6.

11. Voigt K. Ethical concerns in tobacco control nonsmoker and "nonnicotine" hiring policies: The implications of employment restrictions for tobacco control. American Journal of Public Health. 2012;102(11):2013-8.

12. Constance J, Peretti-Watel P. The poor's cigarette. Ethnologie Francaise. 2010;40(3):535-42.

13. Mirza M. Advertising restrictions and market concentration in the cigarette industry: A cross-country analysis. International Journal of Environmental Research and Public Health. 2019;16(18).

14. Gallagher AWA, Gilmore AB, Eads M. Tracking and tracing the tobacco industry: Potential tobacco industry influence over the EU's system for tobacco traceability and security features. Tobacco Control. 2019.

15. Thomas D. L’enfant et l’adolescent, cibles de l’industrie du tabac. Bulletin de l'Académie Nationale de Médecine. 2019;203(7):541-8.

16. McDaniel PA, Forsyth SR. Exploiting the “video game craze”: A case study of the tobacco industry’s use of video games as a marketing tool. PLoS ONE. 2019;14(7).

17. Kim SH, Shanahan J. Stigmatizing smokers: Public sentiment toward cigarette smoking and its relationship to smoking behaviors. Journal of Health Communication. 2003;8(4):343-67.

18. Baha M, Le Faou AL. Smokers' reasons for quitting in an anti-smoking social context. Public Health. 2010;124(4):225-31.

---

## [Decision Letter · Decision Letter 1]

23 Jun 2020

The effect of school smoke-free policies on smoking stigmatization: a European comparison study among adolescents.

PONE-D-20-02662R1

Dear Dr. Robert,

We’re pleased to inform you that your manuscript has been judged scientifically suitable for publication and will be formally accepted for publication once it meets all outstanding technical requirements.

Kind regards,

Stanton A. Glantz

Academic Editor

PLOS ONE

Additional Editor Comments (optional):

Reviewers' comments:

Reviewer's Responses to Questions

**Comments to the Author**

1. If the authors have adequately addressed your comments raised in a previous round of review and you feel that this manuscript is now acceptable for publication, you may indicate that here to bypass the “Comments to the Author” section, enter your conflict of interest statement in the “Confidential to Editor” section, and submit your "Accept" recommendation.

Reviewer #1: All comments have been addressed

Reviewer #2: All comments have been addressed

2. Is the manuscript technically sound, and do the data support the conclusions?

Reviewer #1: Yes

Reviewer #2: Yes

3. Has the statistical analysis been performed appropriately and rigorously? 

Reviewer #1: Yes

Reviewer #2: Yes

4. Have the authors made all data underlying the findings in their manuscript fully available?

Reviewer #1: Yes

Reviewer #2: (No Response)

5. Is the manuscript presented in an intelligible fashion and written in standard English?

Reviewer #1: Yes

Reviewer #2: Yes

6. Review Comments to the Author

Reviewer #1: The authors have addressed my concerns. The revised manuscript is ready for publication in PLOS One.

Reviewer #2: (No Response)

7. PLOS authors have the option to publish the peer review history of their article (what does this mean?). If published, this will include your full peer review and any attached files.

Reviewer #1: No

Reviewer #2: No

---

## [Editor Report · Acceptance letter]

30 Jun 2020

PONE-D-20-02662R1 

The effect of school smoke-free policies on smoking stigmatization: a European comparison study among adolescents. 

Dear Dr. Robert:

I'm pleased to inform you that your manuscript has been deemed suitable for publication in PLOS ONE. Congratulations! Your manuscript is now with our production department. 

Kind regards, 

on behalf of

Professor Stanton A. Glantz 

Academic Editor

PLOS ONE